# A local-global transformer-based model for person re-identification

**Guangjie Liu[1], Ke Xu [1], Jinlong Zhu[1]\*, Yu Ge[1], Xiaoyang Chen[2]**

1 College of Computer Science and Technology, Changchun Normal University, Changchun, Jinlin, China,
2 Jilin Open University, Changchun, Jinlin, China

\* zhujinlong@ccsfu.edu.cn

## Abstract

Person re-identification (ReID) aims to recognize a specific individual across various camera views. State-of-the-art methods have shown that both Transformer-based and CNN-based methods deliver competitive performance. However, Transformer-based methods tend to overlook local features, as they primarily process input sequences holistically, rather than focusing on individual elements or small groups within the sequence. To address this limitation, we introduce an innovative Transformer-based person ReID model that effectively integrates local and global features. The Local Attention Module is added to capture fine-grained features, which are then combined with global features to enhance the model's recognition accuracy. Given the importance of positional information in image data, relative position encoding is incorporated within the Local Attention Module. This encoding method better captures the relative positional relationships between different tokens in an image, thereby improving the model's comprehension of the structural information of the image. Experimental results indicate that the Rank-1 of our model respectively improves by 0.7% and 0.9% on the Market-1501 and DukeMTMC-reID benchmark datasets for person ReID.

## 1 Introduction

Owing to the advancements in deep neural networks and the growing demand for intelligent video surveillance, person re-identification (ReID) has emerged as a significant research domain in computer vision. Person ReID aims to identify or match a specific person across various non-overlapping camera views. The photos taken by different cameras have varying viewpoints, low image resolutions, occlusions, and other factors, which make person ReID a challenging task. The societal impact of robust ReID systems manifests in critical real-world applications that intersect daily life. During public emergencies person ReID enables security teams to rapidly reconstruct movement trajectories across disjointed camera networks, directly expediting rescue operations and threat neutralization. Beyond security, these systems revolutionize urban mobility governance. By anonymously matching pedestrians across

**Data availability statement:** Market-1501 Dataset presented are available from https://www.kaggle.com/datasets/pengcw1/market-1501/dataDukeMTMC-reID. Dataset presented are available from https://www.kaggle.com/datasets/whurobin/dukemtmcreid.

**Funding:** This study was funded by Jilin Provincial Department of science and technology (YDZJ202401376ZYTS). The funders had no role in study design, data collection and analysis, decision to publish, or preparation of the manuscript.

intersections, municipalities optimize public transport routes while preserving citizen privacy. Such anonymized analytics further assist retailers in understanding foot traffic demographics for commercial planning, demonstrating how ReID transforms passive surveillance into proactive, privacy-conscious urban intelligence. It is precisely these high-stakes scenarios demand continual advances in person ReID robustness.

Most models for ReID rely on building and training convolutional neural networks (CNNs) to map each image to a feature representation within the embedding space. CNN-based models such as SOLIDER [1], RGT&RGPR [2], and st-ReID [3] have demonstrated excellent performance on several well-known datasets. However, CNN-based models only focus on small, unrelated regions in the image, which is unfavorable for improving robustness and discriminability. Recently, the emergence of ViT [4] has provided a new alternative. Pure Transformer-based models are comparable to CNN-based models in feature extraction for image recognition. Transformers are capable of modeling the long-range dependencies across the entire image, which differentiates them from CNNs. Pure Transformer-based models such as TransReID [5], LA-Transformer [6], and DiP [7] also attain state-of-the-art (SOTA) performance. Although hybrid CNN-Transformer frameworks attempt to combine local feature extraction with global reasoning, they face inherent constraints that limit their efficacy in ReID. The convolutional inductive bias of CNNs fundamentally conflicts with the permutation-invariant nature of self-attention. This mismatch disrupts feature hierarchy cohesion, causing semantic misalignment between shallow CNN layers and deep Transformer layers.These constraints necessitate a pure Transformer solution.

Pure Transformer-based models generally use self-attention to comprehensively model the global dependencies among elements. However, this approach may neglect local details, as it evenly distributes computing resources to each part of the input data without deeply emphasizing local structures. Consequently, this can lead to limited model performance when dealing with local features. Therefore, a Local Attention(LA) Module is proposed to alleviate this problem. Swin [8] is one of the first visual Transformers based on local self-attention, effectively processing image information through layered and shifting window strategies. The Neighborhood Attention Transformer [9] introduces a variation of self-attention that reduces computational complexity and improves model performance by focusing on the local neighborhood within the input sequence. This model divides the input into overlapping neighborhood partitions and independently performs self-attention within these partitions, thus enhancing performance and increasing processing speed.

On the other hand, self-attention mechanisms in Transformers excel at capturing the intricate relationships among tokens in a sequence. However, they possess an inherent limitation in that they cannot capture the sequence of input tokens. Therefore, it is important for Transformers to incorporate explicit representations of positional information [10]. Kan Wu et al. introduce innovative techniques for relative position encoding tailored specifically for 2D images [11], named image RPE (iRPE), which encode spatial relationships between tokens. Jianlin Su et al. present an innovative approach called Rotary Position Embedding (RoPE) [12] to effectively

leverage positional information. Hence, relative position encoding is introduced to provide relative location information for different tokens.

Due to the inherent limitations of Transformer-based models in processing local details and their inability to capture the sequential order of input tokens, we propose an innovative Transformer-based framework that integrates local and global features for enhanced performance in person ReID. Specifically, using TransReID as a baseline, we introduce a LA Module to capture detailed features of the image. This module partitions the feature map into overlapping blocks, each of which passes through its own relative position encoding self-attention mechanism, and then reassembles the blocks according to their original positions to focus on local features. The LA Module has a smaller receptive field and can pay attention to details such as the patterns on clothes and hats worn by individuals, significantly improving person ReID performance.

The key contributions of this research can be outlined as follows:

1. A novel Transformer-based model that efficaciously integrates both local and global features for person ReID is proposed, emphasizing local details.

2. The LA Module serves as a part of the model crafted to effectively capture detailed local features from input data, which are advantageous for person ReID tasks.

3. Relative position encoding is introduced to enhance the comprehension of the spatial relationships between different tokens, aiding in more precise processing of tokens.

4. Comprehensive experiments indicate that our model attains SOTA performance on classical ReID datasets, including Market-1501 [13] and DukeMTMC-reID [14].

## 2 Related work

### 2.1 Person ReID

CNN-based backbone networks continue to be the predominant method in person ReID. For example, the method proposed by Dengpan Fu et al., which achieves the top rank on the Market-1501 benchmark [15], a classic dataset for person ReID, is founded on ResNet [16]. While CNN-based backbone networks have proven effective, recent research has increasingly focused on Transformer architectures. For instance, the Transformer-based DiP [7] proposed by Dengjie Li et al. has achieved notable rankings.

In the task of person ReID, noisy regions in global features can cause interference, highlighting the need to focus on local features. Local features provide more details and can be generated through pose estimation after pose point localization or by partitioning the image horizontally. Extracting pose points to align poses in different images has proven effective for person ReID tasks [17–20]. Methods such as PCB [21], SAM [22], and MGN [23] extract local features by dividing the image into several approximately horizontal strips. We introduce an innovative approach to enhance local feature extraction by partitioning images into square blocks. This method aims to further refine the representation of local details, thereby enhancing the precision and robustness of person ReID systems.

After extracting features in person ReID, designing loss functions becomes particularly important. In the field of person ReID, the most commonly employed loss functions are cross-entropy loss (ID loss) [24] and triplet loss [25]. The BNNeck method [26] combines them to achieve superior performance.

### 2.2 Vision transformer

Transformers, celebrated for their capacity to grasp long-range dependencies and contextual cues, are increasingly recognized as a promising alternative to traditional CNN-based methods, especially in complex tasks.

Both Transformers and CNNs have their own distinct advantages and disadvantages. Combining these two architectures allows leveraging the strengths of both. Mobile-Former , introduced by Chen Y. et al. [27], combines the advantages of MobileNet and Transformer, serving as a two-way bridge between them and excelling in various classification tasks.

 

Wenshuo Li et al. [28] propose combining Transformers with the CNN-based YOLOv5 to improve performance in target detection tasks. While pure Transformer models are gaining popularity, many approaches to enhance them have been proposed.

The attention module is crucial for Transformers. Introducing novel attention mechanisms can significantly improve Transformer performance. Huaibo Huang et al. [29] propose a simple yet strong super token attention (STA) mechanism, while Haoran You et al. [30] introduce an innovative linear-angular attention mechanism. Xuran Pan et al. [31] propose an innovative LA Module called Slide Attention. These diverse attention mechanisms have variously improved ViT performance, confirming the feasibility of optimizing attention mechanisms to enhance model performance.

Objects in images can appear at various scales. The purpose of multi-scale processing is to capture objects in images at different scales, a technique that is also effective in Transformer models. MPViT [32], proposed by Youngwan Lee et al., constructs a multi-scale structure from fine to coarse, enabling fine-grained and coarse-grained feature representations equivalent, thereby improving performance in various fields of computer vision. Chenglin Yang et al. propose the Lite Vision Transformer (LVT) [33], which adopts different attention mechanisms for fine-grained and coarse-grained features, using Convolutional Self-Attention (CSA) to capture fine-grained features and Recursive Atrous Self-Attention (RASA) to capture coarse-grained features. The success of these studies confirms the necessity of focusing on fine-grained aspects when addressing image-related issues.

The Transformer was originally employed in natural language processing tasks, but since the data volume and computational cost of images are higher than those of text, reducing computational cost is also a research direction for ViT. Lei Zhu et al. propose a novel general vision Transformer, called BiFormer [34], which leverages sparsity to reduce computation. Its performance and computational efficiency have been improved in tasks involving dense predictions. Peixian Chen et al. introduce an innovative decoder-free, Transformer-based (DFFT) object detector [35]. This method reduces object detection to a single-stage, anchor-based dense prediction problem using only an encoder by focusing on two key entrances, thereby improving efficiency. Therefore, when improving the Transformer, it is important to ensure that its complexity does not become excessive.

## 3 Methodology

### 3.1 Overall architecture

For person ReID tasks, it is essential to concentrate on the local details of the image. However, pure Transformer-based models often overlook this aspect. We introduce a LA Module to overcome this limitation. This module splits the feature map into smaller patches based on size and stride, and then applies attention operations. By extracting local features, the LA Module achieves an effective balance between global and local features. Section 3.2 provides the details of the LA Module. Relative position encoding helps the model understand the relative positional relationships between various tokens, which is essential for identifying the positions and spatial arrangements of different objects within an image. Therefore, relative position encoding has been incorporated into the LA Module. Further details are elaborated in Section 3.3.

The LA Module is incorporated into the layers of the Transformer, except for the last one. The input feature map Z is passed into the LA Module to extract local features, resulting in $Z_0'$. The normalized $Z_0'$ is added to the original input feature map Z to obtain the output feature map Z'. The specific method is as follows:

$$Z' = Z + \mathrm{dp}\left(LN\left(Z_0'\right)\right) \tag{1}$$

where Z' is output, Z is input, dp is drop path, and LN is LayerNorm.

The above process is summarized as Local Attention (LA). In a layer of the Transformer, Local Attention is added in the front of it. Significantly, the model is segmented into two components: the global model and the local model, before the

last block. Therefore, no LA Module is included in the final block. The Transformer Block with LA Module is represented as follows:

$$TB = LA + SA + MLP \tag{2}$$

where TB is Transformer Block, LA is Local Attention, and SA is Self-attention.

The whole frame of the model is illustrated in Fig 1. The leftmost column represents the overall architecture, where the input image is divided into tokens after being processed by PatchEmbed. Following PatchEmbed are multiple identical Transformer Blocks. The middle part of the figure shows the specific structure of the Transformer Block. The original Transformer Block is composed of two parts: the self-attention and the MLP. This model adds a LA Module before the attention, incorporating relative position encoding into the LA Module. On the right side of the diagram is the specific structure of the LA Module, where the last attention is RP attention, indicating attention with relative positional encoding.

### 3.2 Local attention module

Attention is a commonly utilized and effective optimization technique. By intensifying the model's emphasis on critical regions within the input data, attention mechanisms can enhance model accuracy and performance, serving a pivotal role in various tasks of computer vision. Self-attention treats each position in the input sequence as a query, weighting and summing the values of other positions by computing the similarity between the query and the key.

Usually, attention mechanisms are used to extract attention from the entire image, often overlooking its finer details. Therefore, a LA Module is added to extract local features while retaining the original attention. The original attention is applied globally to the image. To differentiate between them, the original attention mechanism is called the Global Attention(GA) Module. The output of LA module $Z_0$' is obtained by concatenating the CLS token with local features and then

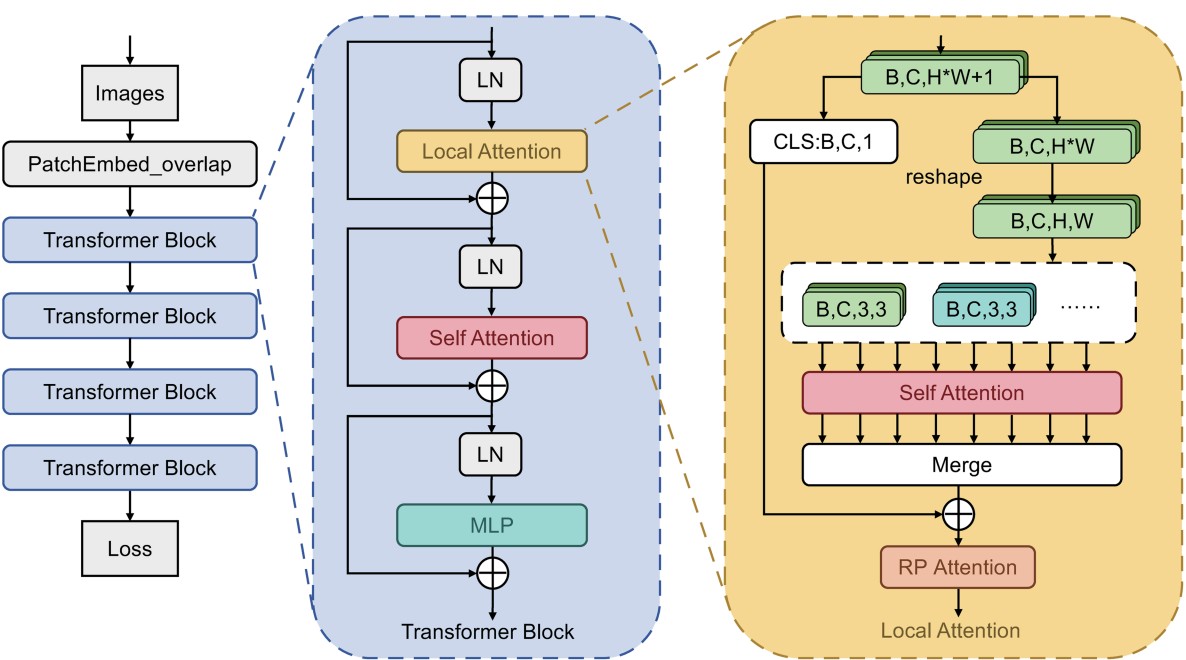

**Fig 1. Framework overview.** The framework is based on TransReID and incorporates two components: the Local Attention Module (LA) and relative position encoding.

passing them through an attention mechanism with relative positional encoding, as follows:

$$Z'_0 = \text{rpAttn}\,(CLS + L) \tag{3}$$

where $Z_0'$ is output feature map, Attn is self-attention with relative positional encoding, and L is local features.

Local features are obtained by combining the input, without the CLS token, into a tensor through pruning and an attention mechanism. The input, without the CLS token, undergoes segmentation and an attention mechanism, and is then merged into a tensor to obtain local features.

$$L = \sum_{i=1}^{k} \text{Attn}\,(\text{Cut}\,(Z_0 - \text{CLS})) \tag{4}$$

where L is local features, $Z_0$ is input, Attn is self-attention mechanism, Cut is dividing regions operation.

Specifically, the dimension size of the input feature map of the LA Module is B×C×(H×W+1), where B denotes batch size, C represents channels, H×W denotes the number of blocks output after PatchEmbed, and 1 represents the CLS token. To extract attention from local partitions, partitions need to be established. First, remove the CLS token to reshape the feature map into B×C×H×W. Then, the feature map is cropped according to size and stride, and the cropped partitions are fed into the self-attention module, which uses relative position encoding. Section 3.3 will provide a detailed description of relative position encoding. When a partition moves beyond the range, the last partition is cropped at the boundary. To preserve the original position of the partition, a new empty tensor of size B×C×H×W is created, allowing the attention-extracted partition to be stacked on the new empty tensor according to its original position. Finally, concatenate this tensor with the original input CLS token, and then apply self-attention to allow the CLS token to interact with the local partition.

### 3.3 Relative position encoding

In Transformer models, the concept of relative position serves a pivotal role, particularly in capturing the nuanced relationships between tokens within a sequence. This is especially relevant in computer vision tasks, where the order of elements can carry significant meaning. Relative position is mathematically expressed as a vector indicating the direction from one position to another within the sequence. This vector benefits the self-attention by helping the model understand the distances between different tokens.

However, calculating the relative position vector for each pair of positions during every self-attention calculation can be computationally expensive. This is because, in a typical sequence, there are many pairs of positions, and computing the relative position vector for each pair requires additional operations. Moreover, in many cases, the relative position vectors between certain pairs of positions are the same, making separate calculations unnecessarily redundant.

To address this issue, a more efficient method is proposed. Before performing the self-attention calculation, a position bias matrix M is computed based on the window size. This matrix essentially pre-calculates the relative position vectors for all possible pairs of positions within the window. When the self-attention calculation requires the relative position bias vectors, instead of computing them on the fly, the model simply selects vectors of the same size from the pre-computed position bias matrix. This approach significantly reduces computational cost by avoiding redundant calculations of the same relative position vectors multiple times. During the calculation of self-attention, a relative position bias is incorporated into each attention head to refine the calculation of similarity among elements. The formula is as follows:

$$\text{Attention}\,(Q, K, V) = \text{softmax}\left(\frac{QK^T}{\sqrt{d_k}} + M\right)V \tag{5}$$

where M represents the relative position vector extracted from the position bias matrix.

 

The position bias matrix is obtained by performing linear transformations on the feature map of the same dimensions. Fig 2 depicts the architecture of relative position self-attention. The input feature map has a shape of B×N×C. It undergoes a linear transformation that produces K, Q, and V matrices of the same shape. The product of Q and the transpose of K is then multiplied by a scaling factor. This step produces the $QK^T$ matrix of shape HN×HN. However, the position bias matrix, which introduces positional information into the attention mechanism, requires additional handling. Since the input feature map includes a CLS token, the position bias matrix must be expanded to accommodate this extra dimension, ensuring it matches the shape of the $QK^T$ matrix. After ensuring the position bias matrix is of the correct shape, it is combined with the $QK^T$ matrix by addition. Ultimately, a softmax function is applied to the attention scores, transforming them into probabilities. These normalized probabilities are then multiplied by the V matrix, which was derived from the input feature map in a similar manner to the Q and K matrices. This step produces the weighted sum of the values, which is the output. This output is then used in subsequent layers of the Transformer, contributing to the model's ability to process sequences and generate predictions or classifications.

## 4 Experiments

This section offers an in-depth overview of the dataset, evaluation metrics, and implementation specifics of the proposed model. Comprehensive ablation studies are performed to validate the efficacy of our model's design. A comparative analysis is conducted to assess our model relative to current SOTA methods using two person ReID datasets.

### 4.1 Dataset and evaluation indicators

We selected two classic datasets in person ReID to evaluate the model: Market-1501 and DukeMTMC-reID. Both datasets include multiple images of each identity, taken from various cameras. Detailed information is shown in the Table 1.

The experimental setup adopts the classic person ReID setting and evaluates using the metrics described in reference [13], which are mean average precision (mAP) and the cumulative matching characteristic curve (CMC). Average precision (AP) is the area under the precision-recall curve for a single category. mAP is calculated by summing the APs

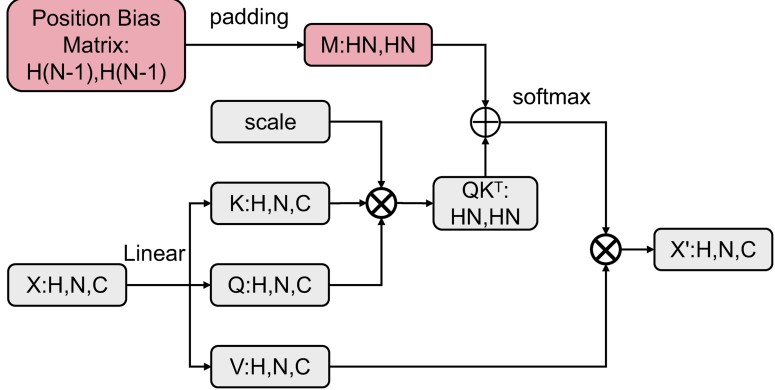

**Fig 2**. **Relative position self-attention structure.**

**Table 1. Detailed statistics of Market-1501 and DukeMTMC-reID.**

| Dataset | Ids | images | cameras |
|---|---|---|---|
| *Market–1501* | 1,501 | 32,668 | 6 |
| *DukeMTMC–reID* | 1,404 | 36,441 | 8 |

of all categories and then taking the average, reflecting the overall recognition accuracy of the model. CMC calculates the ratio of correctly judged labels among the top K individuals in similarity ranking to the total number of tests. Commonly used methods include Rank-1, Rank-5, Rank-10 and mAP.

## 4.2 Implementation

TransReID is selected as the baseline model for our approach.The LA Module is integrated into the layers of the Transformer, except for the last one, with the partition size set to 2×2 and the stride set to 4×4. The image size is adjusted to a fixed resolution of 256×128 for all experiments. The initial learning rate is set to 0.024 with cosine learning rate decay, and training is performed for 120 epochs. The batch size is set to 64. All experiments are conducted using PyTorch on an NVIDIA A100 Tensor Core GPU. Evaluation protocols: Adhering to the conventions in the ReID community, we assess all methods using CMC and mAP.

## 4.3 Ablation experiment

We select TransReID as our baseline and conduct a comprehensive ablation study to evaluate the LA Module's effectiveness across various parameters and structures.

Impact of partition size and stride. The LA Module partitions the input and computes attention within each partition independently, requiring two segmentation parameters: partition size and stride. Each partition is isolated based on its dimensions; features are extracted via self-attention before advancing to the next partition according to the stride. Variations in these parameters significantly influence performance (Fig 3).

Smaller partition sizes improve performance by enabling finer focus on local features critical for distinguishing individuals in person re-identification. This granular extraction enhances recognition of subtle differences.

Conversely, larger strides reduce overlap between adjacent partitions, mitigating feature redundancy and broadening coverage. It is essential for full-body recognition. However, excessively large strides risk losing discriminative local details, degrading accuracy. Optimal stride selection thus balances local detail preservation and global feature coverage.

Through comprehensive ablation studies, the optimal configuration was empirically identified as employing a 2×2 partition size with a 4×4 stride. This design captures fine-grained discriminative regions—such as distinctive accessories and textile patterns—while retaining essential contextual information. Smaller partitions risk excessive fragmentation of the feature space, whereas larger partitions incorporate extraneous background data, thereby diluting salient local features. The 4×4 stride operation minimizes overlap between adjacent regions, enhancing computational efficiency and promoting feature diversity. Simultaneously, the larger stride helps preserve spatial contextual relationships across the feature map. This configuration achieves an optimal equilibrium between local feature precision and global structural coherence, thereby facilitating robust representation learning for person re-identification.

The impact of the internal structure within the LA Module. In the original approach, directly concatenating the CLS token with the tensor containing overlapping partition features creates blank regions in the tensor when the stride exceeds the partition size. Therefore, we conducted an ablation study. By adding the original input features to the tensor with overlapping partitions, we fill these blank regions. The combined tensor then passes through the attention mechanism to update the CLS token. Experimental results indicate that this filling strategy leads to a slight performance degradation, although its mAP remains higher than the baseline's.

Alternatively, the CLS token can be directly concatenated with the tensor containing overlapping partition features and output without the attention mechanism. In this case, the CLS token remains unmodified by the LA module's attention, while the extracted local features are integrated into the feature map. Subsequent changes to the CLS token occur within the GA module. Reducing the use of attention significantly reduces training time. Experimental results show that this strategy improves baseline performance by 0.4% as shown in the Table 2.

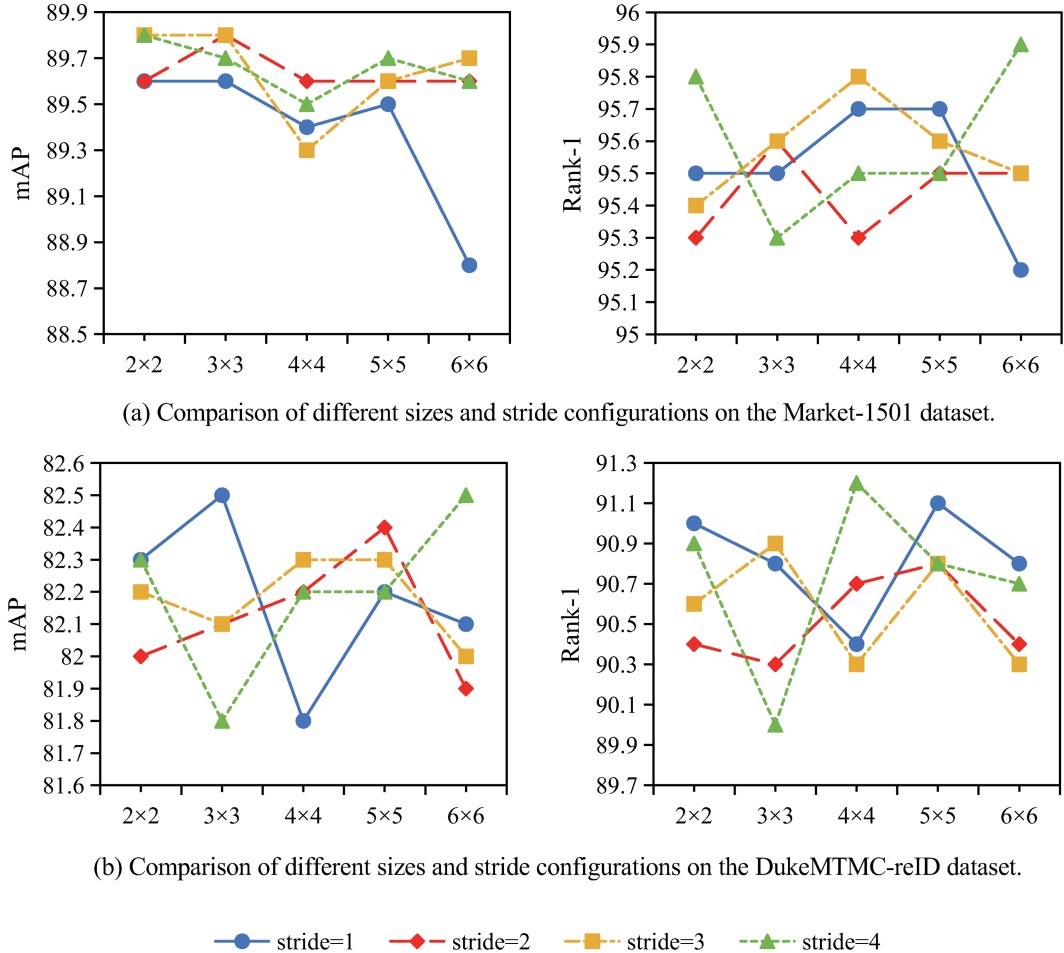

(a) Comparison of different sizes and stride configurations on the Market-1501 dataset.

(b) Comparison of different sizes and stride configurations on the DukeMTMC-reID dataset.

stride=1    stride=2    stride=3    stride=4

**Fig 3. Comparison of various size and stride configurations in the local attention module.**

**Table 2. Comparison of local attention in different module structures.**

| | Market-1501 | | | | DukeMTMC-reID | | | |
|---|---|---|---|---|---|---|---|---|
| *Backbone* | mAP | R1 | R5 | R10 | mAP | R1 | R5 | R10 |
| *Baseline* | 89.0 | 95.1 | 98.3 | 99.1 | 82.2 | 90.0 | 95.7 | 97.1 |
| +*cls* + *attn* | **89.8** | **95.8** | 98.7 | **99.3** | 82.3 | **90.9** | **96.0** | 97.0 |
| +*all* + *attn* | 89.6 | 95.6 | 98.6 | 99.2 | 82.3 | 90.7 | 95.9 | 97.4 |
| +*cls* | 89.4 | 95.3 | 98.7 | 99.2 | 82.3 | 90.5 | 95.8 | 97.2 |
| +*all* | 89.5 | 95.4 | **98.8** | 99.2 | **82.4** | 90.2 | 95.9 | **97.4** |

Similarly, the feature map combined with the original input can also be output directly without attention. In this approach, the CLS token is not updated by the LA module's attention but will be processed along with the feature map by the attention mechanism in the subsequent GA module. Although reducing attention usage saves training time, it degrades model performance. Nevertheless, under this strategy, the model achieves an mAP 0.5% higher than the baseline on Market-1501 as shown in the Table 2.

The impact of the model architecture. The entire model consists of repeated Transformer blocks. In our proposed architecture, the LA Module is positioned at the front of each Transformer block. Alternative configurations for integrating the

LA Module into the Transformer block were also explored. For instance, the LA and GA Modules can be connected separately to the MLP and then combined in series or parallel to form a block structure. Another option is connecting the LA Module directly to the MLP. As shown in the Table 3, experimental results indicate that positioning the LA Module before the GA Module yields optimal performance.

Placing the LA module before the GA block proves more effective than other configurations. This is because the LA module primarily extracts detailed local features, whereas the GA block focuses on capturing global information. When placed first, the LA module processes local features initially, providing richer input for subsequent global feature modeling. This arrangement facilitates more effective integration of local and global features, ultimately enhancing overall recognition performance.

Directly connecting the LA module to the MLP results in the loss of global feature modeling. Although the LA module's attention mechanism partially processes the entire feature map, its performance remains inferior to models incorporating the GA module. The GA module is essential for capturing global features and long-range dependencies, complementing the LA module's focus on local details. While concatenating or combining the modules in parallel could theoretically enhance expressive capacity, in practice, it significantly increases model complexity, leading to substantially higher computational costs and training time, which detrimentally impacts overall performance. Specifically, excessive complexity not only complicates training but may also introduce redundant information, further degrading performance.

The impact of relative position encoding. Our model employs self-attention in two modules: the LA Module and the GA Module. We integrate relative position encoding (RPE) into the self-attention mechanism and explore various methods for incorporating it into these modules. Experiments include applying RPE to both LA and GA modules simultaneously, as well as to each module individually. Results demonstrate that applying RPE solely to the LA module yields the best performance. This is because the LA module focuses on extracting local detail features. Introducing RPE enables self-attention to more precisely capture the relative spatial relationships between these local features, thereby enhancing the model's sensitivity to fine-grained details. In contrast, applying RPE to the GA module did not yield significant improvements. This is likely because the GA module primarily models global information, involving larger regions with fewer local details. Consequently, the impact of RPE on global features is limited, resulting in only marginal performance gains. Under this strategy (RPE only in LA), the model achieves a 0.4% higher mAP than the baseline on Market-1501 and 0.2% higher mAP on DukeMTMC-reID as shown in the Table 4.

**Table 3. Comparison of local attention in different module structures.**

| Backbone | Market-1501 | | | | DukeMTMC-reID | | | |
|---|---|---|---|---|---|---|---|---|
| | mAP | R1 | R5 | R10 | mAP | R1 | R5 | R10 |
| Baseline | 89.0 | 95.1 | 98.3 | 99.1 | 82.2 | 90.0 | 95.7 | **97.1** |
| L + A + MLP | **89.8** | **95.8** | **98.7** | **99.3** | **82.3** | **90.9** | **96.0** | 97.0 |
| LA + MLP | 84.6 | 93.4 | 97.6 | 98.8 | 80.6 | 90.1 | 95.5 | 97.0 |
| series | 83.5 | 92.3 | 97.4 | 98.4 | 79.9 | 89.2 | 95.3 | 96.9 |
| parallel | 80.7 | 91.4 | 97.1 | 98.3 | 74.1 | 85.8 | 93.9 | 95.9 |

**Table 4. Comparison of different methods for adding relative position encoding.**

| Backbone | Market-1501 | | | | DukeMTMC-reID | | | |
|---|---|---|---|---|---|---|---|---|
| | mAP | R1 | R5 | R10 | mAP | R1 | R5 | R10 |
| Baseline | 89.0 | 95.1 | 98.3 | 99.1 | 82.2 | 90.0 | 95.7 | 97.1 |
| +rplocal | **89.4** | **95.2** | **98.6** | **99.1** | **82.4** | **91.0** | **96.1** | **97.6** |
| +rp | 76.7 | 89.9 | 96.5 | 97.6 | 69.0 | 81.8 | 91.5 | 94.2 |
| +local + rp | 77.3 | 89.6 | 96.5 | 97.8 | 67.0 | 80.9 | 91.4 | 93.4 |
| +rplocal + rp | 69.0 | 84.5 | 94.7 | 97.3 | 68.0 | 80.7 | 90.5 | 93.6 |

## 4.4 Qualitative analysis

A qualitative analysis was conducted to provide intuitive insights into the performance enhancements afforded by the proposed local-global module. Fig 4 juxtaposes representative success and failure cases retrieved by our model on the DukeMTMC-reID dataset. The visualization specifically presents the query image alongside the corresponding top-1 matching result from the gallery set. The primary advantage of our method over baseline models lies in its capacity to leverage local patterns including unique textile designs logos or accessories that are often overlooked by models relying solely on global features. In these successful cases as shown in Fig 4(a) the attention mechanism successfully localizes and utilizes these discriminative regions thereby retrieving the correct identity where baseline methods may fail. Conversely the model exhibits limitations when processing queries containing subjects adorned in plain non textured attire that lacks salient local cues as seen in Fig 4(b). In such cases where discriminative information is inherently absent from the imagery our model does not yield a significant improvement over existing approaches. This underscores a persistent challenge within the field and highlights a critical direction for future research aimed at improving Re ID performance under low information conditions.

## 4.5 Comparison with SOTA methods

Finally, as depicted in Table 5, we compare our method with SOTA approaches on the Market-1501 and DukeMTMC-reID datasets. Our model achieves 89.8% and 82.3% mAP on Market-1501 and DukeMTMC-reID, respectively. Overall, our proposed model either outperforms other models or achieves comparable performance, demonstrating its effectiveness.

Our experimental results demonstrate consistent advantages over existing methods. While baseline models struggle with fine-grained discrimination, our approach achieves superior mAP by effectively combining local detail extraction such as accessory patterns, partial garment features with global modeling. This dual focus addresses critical limitations in current SOTA methods.

Results on Market-1501: As illustrated in Table 5, our model demonstrates outstanding performance in terms of mAP, R1, and R5 compared to other SOTA methods. Specifically, compared to TransReID used as the baseline, our model shows an improvement of 0.8% in mAP and 0.7% in R1. Moreover, when compared with TMGF based on ViT, our model achieves a substantial lead with a 5.4% higher mAP and 4.2% higher R1.

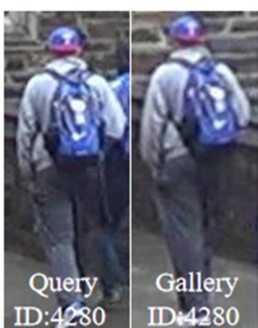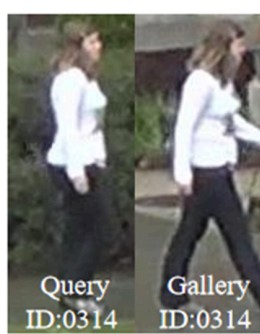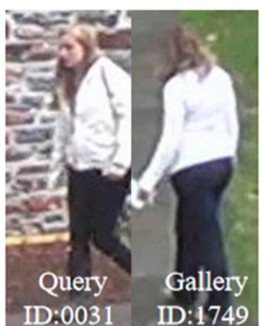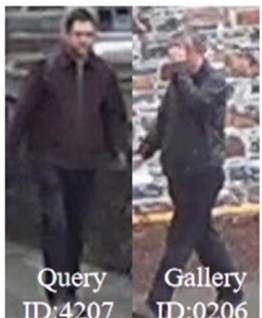

(a) Successful Cases　　　　(b) Failure Cases

**Fig 4**. Qualitative results on DukeMTMC-reID.

**Table 5. Comparison with state-of-the-art methods for person ReID problem.**

| Backbone | Market-1501 | | DukeMTMC-reID | |
|---|---|---|---|---|
| | mAP | R1 | mAP | R1 |
| ReMix [36] | 70.5 | 87.8 | 63.3 | 79.0 |
| TMGF [37] | 89.5 | 95.5 | 76.8 | 86.7 |
| BPB [38] | 87.0 | 95.1 | 78.3 | 89.6 |
| O2CAP [39] | 82.7 | 92.5 | 71.2 | 83.9 |
| DCAL [40] | 87.5 | 94.7 | 80.1 | 89.0 |
| ICE [41] | 82.3 | 93.8 | 69.9 | 83.3 |
| CAL [42] | 89.5 | 95.5 | 80.5 | 90.0 |
| AAformer [43] | 87.7 | 95.4 | 80.0 | 90.1 |
| SAN [22] | 88.0 | **96.1** | 75.5 | 87.9 |
| ISP [44] | 88.6 | 95.3 | 80.0 | 89.6 |
| SONA [45] | 88.8 | 95.6 | 78.3 | 89.4 |
| MGN [23] | 86.9 | 95.7 | 78.4 | 88.7 |
| PCB + RPP [21] | 81.6 | 93.8 | 69.2 | 83.3 |
| PCB [21] | 77.4 | 92.3 | 66.1 | 81.7 |
| IBNNet50a [46] | 88.2 | 95.0 | 79.1 | 90.1 |
| BOT [26] | 85.9 | 94.5 | 76.4 | 86.4 |
| TransReID [5] | 89.0 | 95.1 | 82.2 | 90.0 |
| ours | **89.8** | 95.8 | **82.3** | **90.9** |

Results on DukeMTMC-reID: As shown in Table 5, our model surpasses other SOTA methods in performance. Specifically, compared to TransReID, our model achieves a modest increase of 0.1% in mAP and a notable improvement of 0.9% in R1. Furthermore, compared to CLIP ReID, while our model shows a slight decrease in mAP, it exhibits a significant increase of 0.9% in R1.

Many models also integrate local features into person re-identification, such as PCB and MGN. Compared to PCB, our model's main advantage lies in its ability to simultaneously capture both local and global features. While PCB enhances local feature extraction by dividing the image into multiple parts and processing each part independently, it neglects the global context and relies on fixed partitioning, making it susceptible to pose variations and occlusions. MGN captures local information at different granularities through multi-level feature learning, but its focus remains on local feature extraction, potentially overlooking the effective integration of global information. In contrast, our model introduces a Local Attention Module, which is combined with a Global Attention Module, enabling flexible attention to key local regions in the image and better understanding the spatial relationships between different regions through relative position encoding. This innovation improves the model's adaptability to complex scenarios. Experimental results show that based on this innovative design, our model achieves superior performance.

While the proposed local attention module significantly enhances feature discriminability, it introduces incremental computational complexity relative to the original transformer-based model. This overhead stems from computing additional local attention in parallel to the global attention mechanism.Future work may explore attention distillation or sparse computation to alleviate this cost.

As shown in Fig 5, our model demonstrates strong performance on benchmark datasets for person ReID, outperforming existing methods. This achievement underscores the efficacy of local feature extraction in person ReID tasks. The model achieving competitive results suggests that local approaches can be a reliable alternative to global feature-based methods, providing a promising pathway for future research and development.

## 5 Conclusions

In this paper, we introduce a pure Transformer-based model that accounts for both local and global factors in person ReID tasks, thereby enhancing recognition accuracy. This method adds a LA Module before the traditional self-attention module

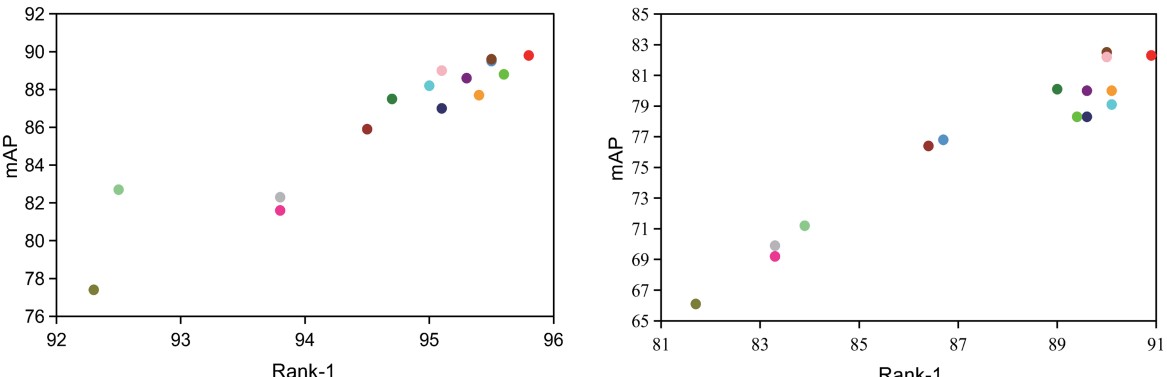

(a) Different Baselines on the Market-1501 Dataset.    (b) Different Baselines on the DukeMTMC-reID Dataset.

● ours ● TMGF ● BPB ● CLIP-ReID ● O2CAP ● DCAL ● ICE ● AAformer ● ISP ● SONA ● PCB+RPP ● PCB ● IBN-Net50-a ● BOT ● TransReID

**Fig 5**. **Performance of Different Baselines on Market-1501 and DukeMTMC-reID Datasets.**

to extract local features, making the model to balance both global and local aspects. Furthermore, attention mechanisms cannot capture the order of input tokens. Even with absolute positional encoding applied, they cannot distinguish between different tokens. Therefore, relative position encoding is integrated into the LA Module to capture positional relationships between different tokens, enabling the model to better understand spatial arrangements and enhance its capability to distinguish fine-grained details. Unlike existing visual Transformers, our method focuses more on image details to achieve a more detailed extraction of person features, thereby improving the model's accuracy. Extensive experiments have shown that our model performs better than its baseline and is more effective compared to current mainstream models in the field of person ReID.

## Author contributions

**Formal analysis:** Yu Ge.

**Methodology:** Guangjie Liu.

**Project administration:** Guangjie Liu.

**Supervision:** Guangjie Liu.

**Validation:** Yu Ge.

**Visualization:** Xiaoyang Chen.

**Writing – original draft:** Ke Xu.

**Writing – review & editing:** Jinlong Zhu.

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
