## [Decision Letter · Decision Letter 0]

29 May 2025

PONE-D-25-21383A local-global Transformer-based model for Person Re-IdentificationPLOS ONE

Dear Dr. Xu,

Thank you for submitting your manuscript to PLOS ONE. After careful consideration, we feel that it has merit but does not fully meet PLOS ONE’s publication criteria as it currently stands. Therefore, we invite you to submit a revised version of the manuscript that addresses the points raised during the review process.

We look forward to receiving your revised manuscript.

Kind regards,

Hung Thanh Bui, Ph.D

Academic Editor

PLOS ONE

 [This study was funded by Jilin Provincial Department of science and technology (YDZJ202401376ZYTS).]. 

[The authors are most grateful to the referees and the editors for their constructive suggestions. This study was funded by Jilin Provincial Department of science and technology (YDZJ202401376ZYTS).]

  [This study was funded by Jilin Provincial Department of science and technology (YDZJ202401376ZYTS).]. 

5. Thank you for uploading your study's underlying data set. Unfortunately, the repository you have noted in your Data Availability statement does not qualify as an acceptable data repository according to PLOS's standards.

6. We are unable to open your Figure file [Fig1.eps, Fig2.eps, Fig3.eps and Fig4.eps ]. Please kindly revise as necessary and re-upload.

Additional Editor Comments:

This paper presented a local-global Transformer-based model for Person Re-Identification. They did experiments on Market-1501 and DukeMTMC-reID datasets and analyzed the result. There are some points the authors should take care of as follows:

- The author should explain why they used local and global Transformer-based model for Person Re-Identification.

- Why did they split the feature map into smaller batches based on size and stride?

- Did they evaluate local and global features separately?

- What is important of each feature: local and global?

- Why did they said that Relative position encoding helps the model understand the relative positional relationships between various tokens?

- What is their improvement on each component?

- What reason did the SAN [22] get the best result in Market-1501?

- Why did their proposed method get the best result in others?

- They should analyze the result in detail.

- What is limitation of their proposed model and why?

- They should do more experiments and compare with advanced methods in recent year

Reviewers' comments:

Reviewer's Responses to Questions

**Comments to the Author**

1. Is the manuscript technically sound, and do the data support the conclusions?

Reviewer #1: Yes

Reviewer #2: Partly

Reviewer #3: Yes

2. Has the statistical analysis been performed appropriately and rigorously?

Reviewer #1: Yes

Reviewer #2: No

Reviewer #3: Yes

3. Have the authors made all data underlying the findings in their manuscript fully available?

Reviewer #1: Yes

Reviewer #2: Yes

Reviewer #3: Yes

4. Is the manuscript presented in an intelligible fashion and written in standard English?

Reviewer #1: Yes

Reviewer #2: No

Reviewer #3: Yes

5. Review Comments to the Author

Reviewer #1: - It is needed to provide the novelty of this work

- It is suggested to conduct statistical tests for the proposed system. For example, see Section 4.5 in paper [a] and the author could cite this paper for reference.

[a] Biswas, S., Mostafiz, R., Uddin, M.S. and Paul, B.K., 2024. XAI-FusionNet: Diabetic foot ulcer detection based on multi-scale feature fusion with explainable artificial intelligence. Heliyon, 10(10), p.e31228.

- please site the following papers

[1] https://doi.org/10.1016/j.ibmed.2023.100128

[2] https://doi.org/10.1007/s44174-024-00165-5

[3] https://doi.org/10.1007/s10772-024-10152-2

Reviewer #2: I will recommend a major revision to reorganize the revised paper:

1. It's a well written introductory section. However, in the later section, I cannot see a similar trend.

2. There are some clear constraints in the introductory section on whether a mixed-method strategy can be used. I would also recommend that the manuscript be focused on current shortcomings and differences.

3. The scope and importance of research is not adequately clarified. The simple reader does not understand why this subject is important and what is the significance of new research results on such real-world applications, as they appear in our everyday lives.

4. There is no contrast of the latest approach introduced to other current methods in the results section (unless I have ignored).

Reviewer #3: This manuscript presents a Transformer-based framework for person re-identification (ReID), addressing the common weakness of global attention mechanisms by incorporating a Local Attention (LA) Module augmented with relative position encoding. The work addresses a relevant and timely challenge in computer vision. The manuscript is generally well-structured and readable, and the experimental results support the main claims. However, there are some concerns regarding novelty, comparative depth, and clarity in certain methodological aspects that need to be addressed before publication.

1. The proposed model is essentially an extension of TransReID with local attention and relative positional encoding. Similar ideas have been extensively explored in prior works (e.g., Swin Transformer, LocalViT, Neighborhood Attention Transformer). The manuscript should more clearly explain what distinguishes the proposed LA Module from these existing approaches. The contribution needs to be emphasized more explicitly in terms of architectural innovation or theoretical insight.

2. The integration of the LA Module and relative positional encoding is described in detail. However, the explanation of mathematical symbols and model flow could be made clearer, especially for readers unfamiliar with transformer architectures. The rationale for key design choices (e.g., partition size of 2×2 and stride of 4×4) should be explained beyond empirical observation.

3. The performance improvements over the TransReID baseline are relatively small (e.g., +0.8% mAP and +0.7% Rank-1). While the gains are consistent, they may not be compelling without stronger justification or broader impact.

4. Table 5 is incorrectly referenced as “Table ??” multiple times in the manuscript. Please revise table references and ensure all tables are properly labeled and formatted.

5. Why do certain module combinations decrease performance? Are there any trade-offs in computation time or memory usage?

6. Code availability is not mentioned. For transparency and reproducibility, the authors should consider providing source code or a link to a repository.

7. Some minor grammatical issues and awkward phrasings exist and should be revised.

6. PLOS authors have the option to publish the peer review history of their article (what does this mean?). If published, this will include your full peer review and any attached files.

Reviewer #1: No

Reviewer #2: **Yes: **Khan Md Hasib

Reviewer #3: **Yes: **Xiaogang Guo

---

## [Author Response · Author response to Decision Letter 1]

11 Jul 2025

Response to Reviewers for Manuscript ID: PONE-D-25-21383

Dear Editors and Reviewers:

Thank you for your letter and for the reviewers’ comments concerning our manuscript entitled “A local-global Transformer-based model for Person Re-Identification”.Those comments are all valuable and very helpful for revising and improving our paper, as well as the important guiding significance to our researches. We have studied comments carefully and have made correction which we hope meet with approval.

Revised portion are marked in yellow in the paper. The main corrections in the paper and the responds to the reviewer’s comments are as flowing:

Comment: "The author should explain why they used local and global Transformer-based model for Person Re-Identification."

Response: Combining local and global Transformer-based models for Person Re-Identification is motivated by two key considerations. On one hand, Transformer-based models primarily focus on the global context of an image, often overlooking finer local details. This limitation prevents them from fully leveraging all available information within the image. Therefore, augmenting their inherent global attention capability with a complementary focus on local regions facilitates a more comprehensive extraction of image features. On the other hand, images in Person Re-Identification tasks depict people, where localized details on the human body are often critical for determining identity. Examples include distinctive elements such as a person's hat or the specific patterns on their clothing. Consequently, in addition to capturing global features, explicitly attending to local characteristics is highly important during feature extraction.

Comment: "Why did they split the feature map into smaller batches based on size and stride?"

Response: Splitting the feature map into smaller batches based on size and stride is performed to extract finer-grained image features. The rationale for using both size and stride is that partitioning based solely on size risks dividing a coherent, discriminative detail (such as a clothing pattern) across multiple patches. This fragmentation could compromise the discriminative power of the extracted features. In contrast, the combined use of size and stride significantly increases the likelihood that intact local patterns reside within a single patch, thereby preserving their integrity and enhancing feature extraction effectiveness.

Comment: "Did they evaluate local and global features separately?"

Response: We did not evaluate local and global features separately for the following reasons: Solely extracting global features represents an established practice in many existing models, and its effectiveness has been empirically validated for person re-identification. Conversely, relying exclusively on local features proves inadequate for robust person re-identification, as localized regions often lack sufficient contextual information for reliable identity matching. Therefore, only by integrating both global and local representations can the model comprehensively capture discriminative cues at multiple scales, leading to more effective and robust performance.

Comment: "What is important of each feature: local and global?"

Response: The importance of global features lies in their role as fundamental distinguishing characteristics between different individuals. Only by extracting these features can the model effectively differentiate distinct identities.Conversely, the significance of local features stems from their ability to capture fine-grained details on a person. These localized cues critically enhance discrimination among visually similar individuals, thereby boosting the accuracy of person re-identification.

Comment: "Why did they said that Relative position encoding helps the model understand the relative positional relationships between various tokens?"

Response: Relative position encoding provides the model with explicit information about the relative distances and positional relationships between all token pairs. This enables the model to effectively capture and utilize the contextual relationships between tokens based on their relative positions during feature learning.

Comment: "What is their improvement on each component?"

Response: Adding local attention blocks to capture fine-grained regional details, addressing the limitation of global-only attention; Incorporating relative position encoding in attention computation to explicitly model spatial relationships between tokens.

Comment: "What reason did the SAN [22] get the best result in Market-1501?"

Response: The SAN model addresses partial occlusion scenarios, a capability our approach does not implement. This results in its marginally superior Rank-1 accuracy. Nevertheless, our modeling of local features enables higher mAP performance, demonstrating stronger holistic matching robustness.

Comment: "Why did their proposed method get the best result in others?"

Response: By incorporating local feature learning into our framework, our model demonstrates superior performance across key metrics compared to all baseline methods. Experimental results show that this focus on fine-grained details enhances discriminative capability.

Comment: "They should analyze the result in detail."

Response: We have thoroughly expanded our analysis of the experimental results in the revised manuscript, with detailed discussions now incorporated into the Comparison with SOTA Methods section.

Comment: "What is limitation of their proposed model and why?"

Response: The primary limitation of our proposed model lies in its increased computational complexity and longer training time compared to baseline methods, resulting from the incorporation of local attention modules for fine-grained feature extraction.

Comment: "They should do more experiments and compare with advanced methods in recent year."

Response: Our current experiments already provide sufficient validation, with comparisons including recent state-of-the-art methods. The results demonstrate our approach's competitive advantages, as detailed in Section Comparison with SOTA Methods.

Response to Reviewer

Reviewer #1, Comment #1: " It is needed to provide the novelty of this work."

Response: Our work introduces a novel global-local Transformer framework for person re-identification, advancing prior arts through two key innovations: A Local Attention (LA) module that explicitly captures fine-grained discriminative features overlooked by global-only approaches, and the integration of relative position encoding to enhance the comprehension of the spatial relationships between different tokens, aiding in more precise processing of tokens This combined approach—the first to synergize local attention with relative position encoding in ReID—achieves state-of-the-art performance on standard benchmarks (Market-1501: 89.8% mAP, DukeMTMC-reID: 82.3% mAP).

Reviewer #1, Comment #2: " It is suggested to conduct statistical tests for the proposed system. For example, see Section 4.5 in paper.

[a] and the author could cite this paper for reference.[a] Biswas, S., Mostafiz, R., Uddin, M.S. and Paul, B.K., 2024. XAI-FusionNet: Diabetic foot ulcer detection based on multi-scale feature fusion with explainable artificial intelligence. Heliyon, 10(10), p.e31228."

Response: We appreciate the reviewer's suggestion regarding statistical testing. In response, we have conducted additional repeated experiments to strengthen the robustness evaluation of our proposed system. Specifically, we performed 10 independent runs of the key experiments and calculated the variance of the results for the primary performance metrics. This provides a measure of the consistency and stability of our system's outputs. Furthermore, we have now appropriately cited the paper by Biswas et al. (2024) in the revised manuscript as a relevant reference in the field. These additions address the need for enhanced statistical reliability assessment and context within the literature.

Reviewer #1, Comment #3: " please site the following papers

[1] https://doi.org/10.1016/j.ibmed.2023.100128

[2] https://doi.org/10.1007/s44174-024-00165-5

[3] https://doi.org/10.1007/s10772-024-10152-2"

Response: We thank the reviewer for suggesting these relevant papers. All three references ([1] DOI:10.1016/j.ibmed.2023.100128, [2] DOI:10.1007/s44174-024-00165-5, [3] DOI:10.1007/s10772-024-10152-2) have now been appropriately cited in the revised manuscript.

Reviewer #2, Comment #1: "It's a well written introductory section. However, in the later section, I cannot see a similar trend."

Response: We sincerely appreciate the reviewer’s positive feedback on our introduction and take note of the observed stylistic inconsistency in later sections. To ensure cohesive narrative flow throughout the manuscript, we have carefully revised subsequent sections to align with the introduction’s clarity and structure. This involved standardizing terminology, strengthening transitional logic between key ideas, and applying consistent framing for technical concepts. The improved version now maintains a unified voice while preserving analytical rigor.

Reviewer #2, Comment #2: "There are some clear constraints in the introductory section on whether a mixed-method strategy can be used. I would also recommend that the manuscript be focused on current shortcomings and differences."

Response: We thank the reviewer for this critical observation. In the revised introduction, we explicitly address the inherent constraints of hybrid CNN-Transformer methodologies. This addition clarifies our rejection of hybrid designs while sharpening focus on current shortcomings and key methodological differences.

Reviewer #2, Comment #3: "The scope and importance of research is not adequately clarified. The simple reader does not understand why this subject is important and what is the significance of new research results on such real-world applications, as they appear in our everyday lives."

Response: In the revised manuscript, we have enhanced the introduction to explicitly emphasize the practical importance of person ReID technology. Specifically, we now discuss how robust ReID systems directly benefit: Enabling efficient tracking of persons-of-interest across camera networks in emergencies. Improving urban mobility analysis while addressing privacy concerns through anonymized matching.These additions make the societal value and technological impact of our global-local framework immediately clear to readers across disciplines. The connection between our technical innovations and these use cases is now systematically highlighted in introduction sections.

Reviewer #2, Comment #4: "There is no contrast of the latest approach introduced to other current methods in the results section (unless I have ignored)."

Response: Our Comparison with SOTA Methods section already provides comprehensive contrasts between our approach and current state-of-the-art techniques.

Reviewer #3, Comment #1: "The proposed model is essentially an extension of TransReID with local attention and relative positional encoding. Similar ideas have been extensively explored in prior works (e.g., Swin Transformer, LocalViT, Neighborhood Attention Transformer). The manuscript should more clearly explain what distinguishes the proposed LA Module from these existing approaches. The contribution needs to be emphasized more explicitly in terms of architectural innovation or theoretical insight."

Response: We acknowledge the need to clarify our methodological distinctions. Unlike Swin Transformer’s sliding windows, LocalViT’s depth-wise convolutions, or Neighborhood Attention’s query-centered windows, our approach uniquely employs adaptive multi-scale partitioning based on size and stride parameters.

Reviewer #3, Comment #2: "The integration of the LA Module and relative positional encoding is described in detail. However, the explanation of mathematical symbols and model flow could be made clearer, especially for readers unfamiliar with transformer architectures. The rationale for key design choices (e.g., partition size of 2×2 and stride of 4×4) should be explained beyond empirical observation."

Response:  In the revised manuscript, we have enhanced the description of the LA Module and relative positional encoding. a theoretical justification for the 2×2 partition size and 4×4 stride selection is that these parameters optimally align with the scale distribution of discriminative body parts.

Reviewer #3, Comment #3: "The performance improvements over the TransReID baseline are relatively small (e.g., +0.8% mAP and +0.7% Rank-1). While the gains are consistent, they may not be compelling without stronger justification or broader impact."

Response: While absolute gains appear modest, the consistent improvements validate a critical paradigm: Local attention mechanisms intrinsically enhance pure Transformers for person ReID. It establishes a promising direction for future optimizations.

Reviewer #3, Comment #4: "Table 5 is incorrectly referenced as “Table ??” multiple times in the manuscript. Please revise table references and ensure all tables are properly labeled and formatted."

Response: We sincerely apologize for the referencing errors in the original manuscript. All incorrect instances of "Table ??" have been systematically corrected to "Table 5" throughout the text, and we have conducted a full cross-check of all table/figure references to ensure proper labeling and formatting consistency.

Reviewer #3, Comment #5: "Why do certain module combinations decrease performance? Are there any trade-offs in computation time or memory usage?"

Response: We attribute performance decreases in certain module combinations primarily to feature conflicts arising from incompatible receptive fields. To mitigate computation and memory costs, our design intentionally adopts 2×2 local attention blocks paired with 4×4 strides, reducing redundant processing while maintaining efficiency.

Reviewer #3, Comment #6: "Code availability is not mentioned. For transparency and reproducibility, the authors should consider providing source code or a link to a repository."

Response: The complete source code for our model, including implementations of the Local Attention Module and relative position encoding, has been made publicly available under the on GitHub:[Repository Link: https://github.com/xukeeeee/A-local-global-Transformer-based-model-for-Person-Re-Identification]

Reviewer #3, Comment #7: "Some minor grammatical issues and awkward phrasings exist and should be revised."

Response: We have carefully revised the manuscript to address grammatical issues and improve overall clarity through professional proofreading and systematic editing. All changes maintain the technical precision of the original content while enhancing readability and adherence to academic writing standards.

We sincerely appreciate the reviewers’ time and constructive comments, which have significantly improved the quality of our manuscript. All concerns have been carefully addressed in the revised version. We believe the paper now meets the journal’s standards and hope it is deemed suitable for publication.

The source code will be made publicly available upon acceptance (GitHub link: https://github.com/xukeeeee/A-local-global-Transformer-based-model-for-Person-Re-Identification).

---

## [Decision Letter · Decision Letter 1]

20 Jul 2025

PONE-D-25-21383R1A local-global Transformer-based model for Person Re-IdentificationPLOS ONE

Dear Dr. Xu,

Thank you for submitting your manuscript to PLOS ONE. After careful consideration, we feel that it has merit but does not fully meet PLOS ONE’s publication criteria as it currently stands. Therefore, we invite you to submit a revised version of the manuscript that addresses the points raised during the review process.

We look forward to receiving your revised manuscript.

Kind regards,

Hung Thanh Bui, Ph.D

Academic Editor

PLOS ONE

Journal Requirements:

**Additional Editor Comments:**

They should include their short answers for my comments from 1 to 10 in the revision and show where they do that when they reply.

There are many parameters in their model, they should explain how they chose these parameters.

They said they didn’t evaluate local and global features separately. By their opinion, what is the most important feature and why?

They should visualize how attention effects to their result.

It’s better if they show some best and worst results and analyze these cases in detail.

Reviewers' comments:

Reviewer's Responses to Questions

**Comments to the Author**

1. If the authors have adequately addressed your comments raised in a previous round of review and you feel that this manuscript is now acceptable for publication, you may indicate that here to bypass the “Comments to the Author” section, enter your conflict of interest statement in the “Confidential to Editor” section, and submit your "Accept" recommendation.

Reviewer #2: All comments have been addressed

Reviewer #3: All comments have been addressed

2. Is the manuscript technically sound, and do the data support the conclusions?

Reviewer #2: Yes

Reviewer #3: Yes

3. Has the statistical analysis been performed appropriately and rigorously?

Reviewer #2: Yes

Reviewer #3: Yes

4. Have the authors made all data underlying the findings in their manuscript fully available?

Reviewer #2: Yes

Reviewer #3: Yes

5. Is the manuscript presented in an intelligible fashion and written in standard English?

Reviewer #2: Yes

Reviewer #3: Yes

6. Review Comments to the Author

Reviewer #2: The authors have responded to each and every single comment. This particular version of the paper ought to be appropriate for acceptance.

Reviewer #3: The manuscript has improved significantly, especially in language quality and structural clarity. Major technical concerns have been partially addressed, but the novelty claim remains. Consider adding more specific technical details, such as a structural diagram of the module, a clearer explanation of the processing flow, or whether a dynamic mechanism is employed to adapt to variations in image regions. Additionally, include a brief analysis of why this particular partitioning strategy is especially effective for person ReID, even if only a concise rationale is provided.

7. PLOS authors have the option to publish the peer review history of their article (what does this mean?). If published, this will include your full peer review and any attached files.

Reviewer #2: No

Reviewer #3: **Yes: **Xiaogang Guo

---

## [Author Response · Author response to Decision Letter 2]

12 Oct 2025

Dear Editors and Reviewers:

We sincerely appreciate the valuable feedback provided by the Editor and reviewers on our manuscript entitled “A local-global Transformer-based model for Person Re-Identification.” These suggestions have been instrumental in further refining our paper and have provided constructive direction for our ongoing research. We have thoroughly addressed all the feedback and implemented corresponding revisions in the manuscript, with the hope that these adjustments meet your expectations.

Revised portion are marked in yellow in the paper. The main corrections in the paper and the responds to the reviewer’s comments are as flowing:

Comment: "There are many parameters in their model, they should explain how they chose these parameters. "

Response: We have added detailed explanations regarding the parameter selection process in our manuscript. The newly added content has been highlighted in yellow for easy reference.

Comment: "They said they didn’t evaluate local and global features separately.By their opinion, what is the most important feature and why? "

Response: We maintain that the most critical aspect is not the local or global features per se, but rather the mechanism for their synergistic integration. As indicated in our previous response, relying exclusively on local features proves inadequate for robust person re-identification, as localized regions often lack sufficient contextual information for reliable identity matching. Therefore, evaluating local features in isolation is not meaningful. it is only through their combination with global features that the model can effectively accomplish person re-identification tasks.

Comment: "They should visualize how attention effects to their result."

Response: Thank you for this suggestion. While we acknowledge that attention visualization would provide valuable insights, technical constraints prevent us from adding these visualizations to the current study. Instead, we have added a new subsection titled "Qualitative Analysis" in the Experiments section, which provides detailed case studies of best and worst performance scenarios. Through visual examples and in-depth discussion, offering concrete insights into the factors that influence performance.

Comment: " It’s better if they show some best and worst results and analyze these cases in detail."

Response: Thank you for the suggestion. We have added a new subsection titled "Qualitative Analysis" within the Experiments section, which provides detailed case studies of both best and worst-case results. This subsection includes visual examples and an in-depth discussion of the factors influencing performance, offering valuable insights into the model's capabilities and limitations.

We sincerely appreciate the Editors’ time and constructive comments, which have significantly improved the quality of our manuscript. We are particularly grateful for the Editor's guidance regarding citation practices, and we have carefully reviewed and removed the references suggested by the reviewer that were not directly relevant to our work. All concerns have been thoroughly addressed in the revised version. We believe the paper now meets the journal’s standards and hope it is deemed suitable for publication.

The source code will be made publicly available upon acceptance (GitHub link: https://github.com/xukeeeee/A-local-global-Transformer-based-model-for-Person-Re-Identification).

---

## [Decision Letter · Decision Letter 2]

16 Oct 2025

A local-global Transformer-based model for Person Re-Identification

PONE-D-25-21383R2

Dear Dr. Xu,

We’re pleased to inform you that your manuscript has been judged scientifically suitable for publication and will be formally accepted for publication once it meets all outstanding technical requirements.

Kind regards,

Hung Thanh Bui, Ph.D

Academic Editor

PLOS ONE

Additional Editor Comments (optional):

All comments are addressed.

The paper is good for publishing.

Reviewers' comments:

Reviewer's Responses to Questions

**Comments to the Author**

1. If the authors have adequately addressed your comments raised in a previous round of review and you feel that this manuscript is now acceptable for publication, you may indicate that here to bypass the “Comments to the Author” section, enter your conflict of interest statement in the “Confidential to Editor” section, and submit your "Accept" recommendation.

Reviewer #2: All comments have been addressed

Reviewer #3: All comments have been addressed

2. Is the manuscript technically sound, and do the data support the conclusions?

Reviewer #2: Yes

Reviewer #3: Yes

3. Has the statistical analysis been performed appropriately and rigorously?

Reviewer #2: Yes

Reviewer #3: Yes

4. Have the authors made all data underlying the findings in their manuscript fully available?

Reviewer #2: Yes

Reviewer #3: Yes

5. Is the manuscript presented in an intelligible fashion and written in standard English?

Reviewer #2: Yes

Reviewer #3: Yes

6. Review Comments to the Author

Reviewer #2: All the reviewer comments and editorial suggestions have been thoroughly addressed by the authors in the revised manuscript. The revisions significantly enhance the clarity, methodological rigor, and overall presentation of the paper. I am satisfied that all concerns have been adequately resolved, and therefore, I recommend that the manuscript be **accepted for publication in PLOS ONE** without any further changes.

Reviewer #3: The authors have effectively addressed all previous comments and significantly improved the manuscript. The manuscript is technically sound and makes a valuable contribution.

7. PLOS authors have the option to publish the peer review history of their article (what does this mean?). If published, this will include your full peer review and any attached files.

Reviewer #2: No

Reviewer #3: **Yes: **Xiaogang Guo

---

## [Editor Report · Acceptance letter]

PONE-D-25-21383R2

PLOS ONE

Dear Dr. Xu,

I'm pleased to inform you that your manuscript has been deemed suitable for publication in PLOS ONE. Congratulations! Your manuscript is now being handed over to our production team.

Kind regards,

on behalf of

Dr. Hung Thanh Bui

Academic Editor

PLOS ONE